# AD-DROP: Attribution-Driven Dropout for Robust Language Model Fine-Tuning

**Tao Yang[1], Jinghao Deng[1], Xiaojun Quan[1]*, Qifan Wang[2], Shaoliang Nie[2]**

[1]School of Computer Science and Engineering, Sun Yat-sen University [2]Meta AI

[1]{yangt225,dengjh27}@mail2.sysu.edu.cn, quanxj3@mail.sysu.edu.cn

[2]{wqfcr, snie}@fb.com

## Abstract

Fine-tuning large pre-trained language models on downstream tasks is apt to suffer from overfitting when limited training data is available. While dropout proves to be an effective antidote by randomly dropping a proportion of units, existing research has not examined its effect on the self-attention mechanism. In this paper, we investigate this problem through self-attention attribution and find that dropping attention positions with low attribution scores can accelerate training and increase the risk of overfitting. Motivated by this observation, we propose Attribution-Driven Dropout (AD-DROP), which randomly discards some high-attribution positions to encourage the model to make predictions by relying more on low-attribution positions to reduce overfitting. We also develop a cross-tuning strategy to alternate fine-tuning and AD-DROP to avoid dropping high-attribution positions excessively. Extensive experiments on various benchmarks show that AD-DROP yields consistent improvements over baselines. Analysis further confirms that AD-DROP serves as a strategic regularizer to prevent overfitting during fine-tuning.

## 1 Introduction

Pre-training large language models (PrLMs) on massive unlabeled corpora and fine-tuning them on downstream tasks has become a new paradigm [1–3]. Their success can be partly attributed to the self-attention mechanism [4], yet these self-attention networks are often redundant [5, 6] and tend to cause overfitting when fine-tuned on downstream tasks due to the mismatch between their overparameterization and the limited annotated data [7–13]. To address this issue, various regularization techniques such as data augmentation [14, 15], adversarial training [16, 17]), and dropout-based methods [11, 13, 18] have been developed. Among them, dropout-based methods are widely adopted for their simplicity and effectiveness. Dropout [19], which randomly discards a proportion of units, is at the core of dropout-based methods. Recently, several variants of dropout have been proposed, such as Concrete Dropout [20], DropBlock [21], and AutoDropout [22]. However, these variants generally follow the vanilla dropout to randomly drop units during training and pay little attention to the effect of dropout on self-attention. In this paper, we seek to fill this gap from the perspective of self-attention attribution [23] and aim to reduce overfitting when fine-tuning PrLMs.

Attribution [24] is an interpretability method that attributes model predictions to input features via saliency measures such as gradient [25, 26]. It is also used to explain the influence patterns of self-attention in recent literature [23, 27, 28]. Our prior experiment of self-attention attribution (Section 2.2) reveals that attention positions are not equally important in preventing overfitting, and dropping low-attribution positions is more likely to cause overfitting than discarding high-attribution positions. This observation suggests that attention positions should not be treated the same in dropout.

---

*Corresponding author.

Motivated by the above, we propose **A**ttribution-**D**riven **Drop**out (AD-DROP) to better fine-tune PrLMs based on self-attention attribution. The general idea of AD-DROP is to drop a set of self-attention positions with high attribution scores. We illustrate the difference between vanilla dropout and AD-DROP by their attention maps in Figure 1. When fine-tuning a PrLM on a batch of training samples, AD-DROP involves four steps. First, predictions are made through a forward computation without dropping any attention position. Second, we compute the attribution score of each attention position by gradient [25] or integrated gradient [26] attribution methods. Third, we sample a set of positions with high attribution scores and generate a mask matrix for each attention map. Finally, the mask matrices are applied

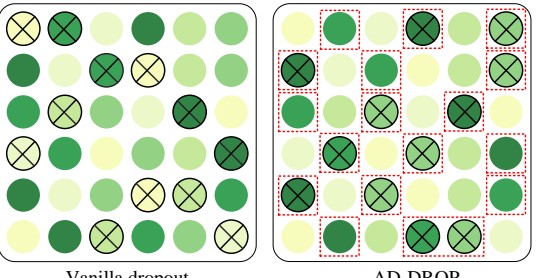

Figure 1: Attention maps of vanilla dropout and our AD-DROP. Darker attention positions indicate higher attribution scores, and crossed circles mean dropped attention positions. Red-dotted boxes refer to candidate discard regions with high attribution scores. Unlike vanilla dropout which randomly discards attention positions, AD-DROP focuses on dropping high-attribution positions in candidate discard regions.

to the next forward computation to make predictions for backpropagation. AD-DROP can be regarded as a strategic dropout regularizer that forces the model to make predictions by relying more on low-attribution positions to reduce overfitting. Nevertheless, excessive neglect of high-attribution positions would leave insufficient information for training. Hence, we further propose a cross-tuning strategy that performs fine-tuning and AD-DROP alternately to improve the training stability.

To verify the effectiveness of AD-DROP, we conduct extensive experiments with different PrLMs (i.e., BERT [1], RoBERTa [2], ELECTRA [29], and OPUS-MT [30]) on various datasets (i.e., GLUE [31], CoNLL-2003 [32], WMT 2016 EN-RO and TR-EN [33], HANS [34], and PAWS-X [35]). Experimental results show that the models tuned with AD-DROP obtain remarkable improvements over that tuned with the original fine-tuning approach. For example, on the GLUE benchmark, BERT achieves an average improvement of 1.98/0.87 points on the dev/test sets while RoBERTa achieves an average improvement of 1.29/0.62 points. Moreover, ablation studies and analysis demonstrate that gradient-based attribution [25, 26] is a more suitable saliency measure for implementing AD-DROP than directly using attention weights or simple random sampling. Moreover, they also demonstrate that the cross-tuning strategy plays a crucial role in improving training stability.

To sum up, this work reveals that self-attention positions are not equally important for dropout when fine-tuning PrLMs. Arguably, low-attribution positions are more difficult to optimize than high-attribution positions, and dropping these positions tends not to relieve but accelerate overfitting. This leads to a novel dropout regularizer, AD-DROP, driven by self-attention attribution. Although proposed for self-attention units, AD-DROP can be potentially extended to other units as dropout.

## 2 Methodology

### 2.1 Preliminaries

Since Transformers [4] are the backbone of PrLMs, we first review the details of self-attention in Transformers and self-attention attribution [23]. Let $\mathbf{X} \in \mathbb{R}^{n \times d}$ be the input of a Transformer block, where $n$ is the sequence length and $d$ is the embedding size. Self-attention in this block first maps $\mathbf{X}$ into three matrices $\mathbf{Q}_h$, $\mathbf{K}_h$ and $\mathbf{V}_h$ via linear projections as query, key, and value respectively for the $h$-th head. Then, the attention output of this head is calculated as:

$$\text{Attention}\left(\mathbf{Q}_h, \mathbf{K}_h, \mathbf{V}_h\right) = \mathbf{A}_h \mathbf{V}_h = \text{softmax}\left(\frac{\mathbf{Q}_h \mathbf{K}_h^{\text{T}}}{\sqrt{d_k}} + \mathbf{M}_h\right) \mathbf{V}_h, \qquad (1)$$

where $\sqrt{d_k}$ is a scaling factor. $\mathbf{M}_h$ is the mask matrix to apply dropout in self-attention, and elements in $\mathbf{M}_h$ will be $-\infty$ if the corresponding positions in attention maps are masked and 0 otherwise.

Based on the attention maps $\mathbf{A} = [\mathbf{A}_1, \mathbf{A}_2, \cdots, \mathbf{A}_H]$ for $H$ attention heads, gradient attribution [25, 36] directly produces an attribution matrix $\mathbf{B_h}$ by computing the following partial derivative:

$$\mathbf{B}_h = \frac{\partial F_c(\mathbf{A})}{\partial \mathbf{A}_h}, \tag{2}$$

where $F_c(\cdot)$ denotes the logit output of the Transformer for class $c$.

To provide a theoretically more sound attribution method, Sundararajan et al. [26] propose integrated gradient, which is employed by Hao et al. [23] as a saliency measure for self-attention attribution. Specifically, Hao et al. [23] compute the attribution matrix $\mathbf{B}_h$ as:

$$\mathbf{B}_h = \frac{\mathbf{A}_h}{m} \odot \sum_{k=1}^{m} \frac{\partial F_c\left(\frac{k}{m}\mathbf{A}\right)}{\partial \mathbf{A}_h}, \tag{3}$$

where $m$ is the number of steps for approximating the integration in integrated gradient, and $\odot$ is the element-wise multiplication operator. Despite its theoretical advantage over gradient attribution, integrated gradient requires $m$ times more computational effort, which is especially expensive when it is applied to all the attention heads in Transformers. Moreover, our experiments in Section 3.4 show that gradient attribution achieves comparable performance with integrated gradient but requires much less computational cost, suggesting that gradient attribution is more desirable for AD-DROP.

## 2.2 A Prior Attribution Experiment

To better motivate our work, we first conduct a prior experiment on MRPC [37] to investigate how different positions in self-attention maps affect fine-tuning performance based on attribution results. RoBERTa_base [2] is used as the base model. To begin with, we first perform a forward computation of the model on each batch of training samples to obtain the logit output of each sample corresponding to the gold label. Then, we obtain an attribution matrix $\mathbf{B}_h$ for the self-attention positions in the first layer[2] by gradient attribution with Eq. (2)

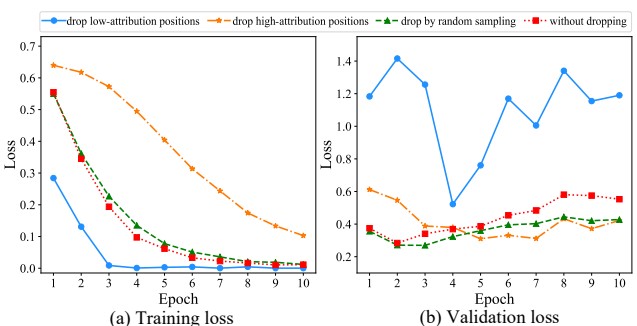

(a) Training loss    (b) Validation loss

Figure 2: Results of training and validation losses when fine-tuning RoBERTa with different dropping strategies on MRPC. The dropping rate is set to 0.3 if it applies.

and sort each row of the matrix. Finally, we sample a set of self-attention positions with high or low attribution scores in each row to generate a mask matrix $\mathbf{M}_h$, which is fed into Eq. (1) to make the final predictions. After each epoch of training, we evaluate the model on the development set. Two baseline dropping strategies (i.e., dropping by random sampling and without dropping any position) are employed for comparison. We plot the loss curves of the model with these dropping strategies on both training and development sets in Figure 2. The observations are threefold. First, dropping low-attribution positions makes the model fit the training data rapidly, whereas it performs poorly on the development set, indicating that the model is not properly trained. Second, compared with the other dropping strategies, dropping high-attribution positions reduces the fitting speed significantly. Third, random dropping only slightly reduces overfitting, compared to the training without dropping. These observations suggest that attention positions are of different importance in preventing overfitting. We conjecture that low-attribution positions are more difficult to optimize than high-attribution positions. While dropping low-attribution positions tends to accelerate overfitting, discarding high-attribution positions helps reduce overfitting.

## 2.3 Attribution-Driven Dropout

Inspired by the observations in Section 2.2, we propose a novel regularizer, AD-DROP, to better prevent overfitting when adapting PrLMs to downstream tasks. The motivation of AD-DROP is to minimize the over-reliance of these models on particular features which may affect their generalization.

---

[2]We provide more results and discussions in Appendix B.

Formally, given a training set $\mathcal{D} = \{(x_i, y_i)\}_{i=1}^{N}$ of $N$ samples, where $x_i$ is the $i$-th sample and $y_i$ is its label, the goal of AD-DROP is to fine-tune a PrLM $F(\cdot)$ of $L$ layers on $\mathcal{D}$. Same as the vanilla dropout [19], AD-DROP is only applied in the training phase.

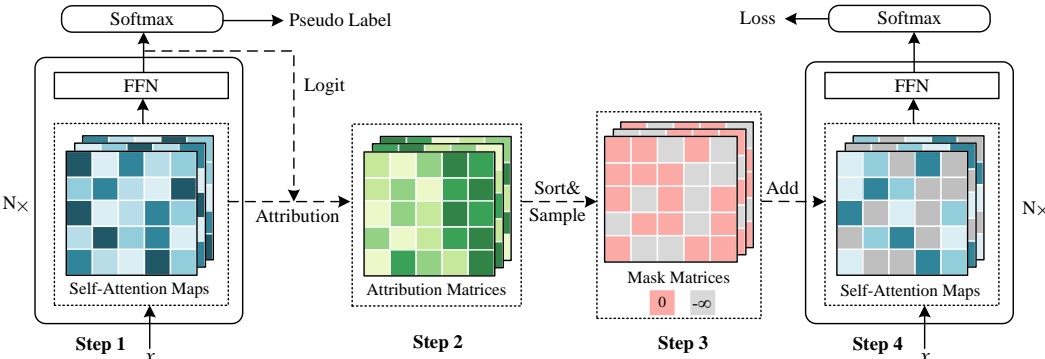

Figure 3: Illustration of AD-DROP in four steps. (1) Conduct the first forward computation to obtain pseudo label $\tilde{c}$. (2) Generate attribution matrices $\mathbf{B}$ via computing the gradient of logit output $F_{\tilde{c}}(\mathbf{A})$ with respect to each attention head. (3) Sort $\mathbf{B}$ and strategically drop some positions to produce mask matrices $\mathbf{M}$. (4) Feed $\mathbf{M}$ into the next forward computation to compute the final loss.

As shown in Figure 3, the idea of AD-DROP can be described in four steps. First, we conduct a forward computation of the model to obtain the label with the highest probability as the pseudo label. The reason we adopt pseudo labels rather than gold labels for attribution will be explained shortly. Specifically, for the input $x_i$ with $n$ tokens, we apply $F(\cdot)$ to encode it and obtain its pseudo label $\tilde{c}$:

$$\tilde{c} = \arg\max_{c} \left( P_F(c|x_i) \right), \tag{4}$$

where $P_F(c|x_i)$ is the probability of class $c$ for $x_i$. After the forward computation, we also obtain a set of attention maps $\mathbf{A} = [\mathbf{A}_1, \mathbf{A}_2, \cdots, \mathbf{A}_H]$ for each layer according to Eq. (1).

Second, we compute the attribution matrices $\mathbf{B} = [\mathbf{B}_1, \mathbf{B}_2, \cdots, \mathbf{B}_H]$ for $H$ heads according to Eq. (2). Specifically, the attribution matrix $\mathbf{B}_h$ for the $h$-th head is computed as:

$$\mathbf{B}_h = \frac{\partial F_{\tilde{c}}(\mathbf{A})}{\partial \mathbf{A}_h}, \tag{5}$$

where $F_{\tilde{c}}(\mathbf{A})$ is the logit output of pseudo label $\tilde{c}$ before softmax.[3]

Third, we generate a mask matrix $\mathbf{M}_h$ based on $\mathbf{B}_h$. To this end, we first sort each row of $\mathbf{B}_h$ in ascending order and obtain a sorted attribution matrix $\widehat{\mathbf{B}}_h$. Then, we define a candidate discard region $\mathbf{S}_h$, in which each element $s_{i,j}$ is defined as:

$$s_{i,j} = \begin{cases} 1, & b_{i,j} < \widehat{b}_{i,\text{int}(n(1-p))} \\ 0, & \text{otherwise} \end{cases} \tag{6}$$

where $b_{i,j}$ and $\widehat{b}_{i,j}$ are elements of $\mathbf{B}_h$ and $\widehat{\mathbf{B}}_h$, respectively, $\text{int}(\cdot)$ is an integer function, and $p \in (0, 1)$ is used to control the size of the candidate discard region. Next, we apply dropout in the region to produce the mask matrix $\mathbf{M}_h$ as:

$$m_{i,j} = \begin{cases} -\infty, & (s_{i,j} + u_{i,j}) = 0 \\ 0, & \text{otherwise} \end{cases} \tag{7}$$

where $u_{i,j} \sim \text{Bernoulli}(1 - q)$ is an element of matrix $\mathbf{U}_h \in \mathbb{R}^{n \times n}$, and $q$ is the dropout rate.

Finally, $\mathbf{M}_h$ is fed into self-attention of Eq. (1) for the second forward computation, and the final output is used to calculate the loss for backpropagation.

---

[3]The negative loss will be used for both regression and token-level tasks, as introduced in Appendix A.

**Discussion** The reasons that AD-DROP uses pseudo labels for attribution are twofold. First, adopting gold labels will divulge label information and lead to inconsistency between training and inference. Second, for misclassified samples in the first forward computation, AD-DROP with gold labels tends to continue to make incorrect predictions because high-attribution attention positions derived from gold labels may be located in low-attribution regions derived from pseudo labels. Therefore, dropping these positions does not help the model correct wrong predictions, while AD-DROP with pseudo labels urges the model to rely on important features in the current pass and may correct the wrong predictions. The attribution with gold labels will be investigated in Section 3.4.

## 2.4 Cross-Tuning Algorithm

We further design a cross-tuning algorithm to avoid dropping high-attribution positions excessively when applying AD-DROP. The idea of cross-tuning is to execute the original fine-tuning and AD-DROP alternately. Specifically, it performs the original fine-tuning at odd epochs and AD-DROP at even epochs. The overall process of cross-tuning is described in Algorithm 1, where Lines 3-5 are the original fine-tuning operations and Lines 7-9 describe the process of AD-DROP.

---

**Algorithm 1** Cross-tuning

---

**Input:** shuffled training samples $\mathcal{D} = \{(x_i, y_i)\}_{i=1}^N$, PrLM $F$ with parameters $\mathbf{W}$
**Output:** updated parameters $\widetilde{\mathbf{W}}$

1: Initialize $F$ with $\mathbf{W}$, $epoch = 1$
2: **while** not converged **do**
3:    Calculate the prediction $P_F(y_i|x_i)$ and loss via forward computation.
4:    **if** $epoch\%2 == 1$ **then**
5:       Backpropagate the loss to update model parameters $\mathbf{W}$.
6:    **else**
7:       Perform AD-DROP by Eq. (4)-(7) to obtain mask matrices $\mathbf{M} = [\mathbf{M}_1, \mathbf{M}_2, \cdots, \mathbf{M}_H]$.
8:       Calculate the new prediction $P_F(y_i|x_i)$ and new loss by feeding $\mathbf{M}$ into Eq. (1).
9:       Backpropagate the new loss to update model parameters $\mathbf{W}$.
10:    $epoch = epoch + 1$
11: **return** $\widetilde{\mathbf{W}} = \mathbf{W}$

---

# 3 Experiments

## 3.1 Datasets

We conduct our main experiments on eight tasks of the GLUE benchmark [31], including SST-2 [38], MNLI [39], QNLI [40], QQP [41], CoLA [42], STS-B [43], MRPC [37], and RTE [44]. The evaluation metrics are Matthew's Corrcoef (Mcc) [45] for CoLA, Pearson Corrcoef (Pcc) [46] for STS-B, and Accuracy (Acc) for the others. To demonstrate that AD-DROP applies to token-level tasks as well, we conduct experiments on Named Entity Recognition (CoNLL-2003 [32]) and Machine Translation (WMT 2016 [33]) datasets, the results of which are shown in Appendix A.2. Besides, we also evaluate AD-DROP on two out-of-distribution (OOD) datasets, including HANS [34] and PAWS-X [35]. The details of these datasets are introduced in Appendix C.1.

## 3.2 Implementation Details

We implement our AD-DROP in Pytorch with the Transformers package [47]. We train the selected PrLMs on GeForce RTX 3090 GPUs. We tune the learning rate in {1e-5, 2e-5, 3e-5} and the batch size in {16, 32, 64}. Following Miao et al. [17], we perform early stopping to choose the number of training epochs on GLUE. The two critical hyperparameters $p$ and $q$ are searched within $[0.1, 0.9]$ with step size 0.1. For integrated gradient in Eq. (3), we follow Hao et al. [23] and set $m$ to 20. We apply AD-DROP only in the first layer for the datasets of SST-2, MNLI, QNLI, QQP, and STS-B since the fine-tuning on these datasets is stable and less likely to cause overfitting. For the rest datasets, we apply AD-DROP in all layers. We provide the detailed hyperparameter settings on each dataset in Appendix C.2. Our code is available at `https://github.com/TaoYang225/AD-DROP`.

Table 1: Overall results of fine-tuned models on the GLUE benchmark. The symbol † denotes results directly taken from the original papers. The best average results are shown in bold.

| Methods | SST-2 | MNLI | QNLI | QQP | CoLA | STS-B | MRPC | RTE | Average |
|---|---|---|---|---|---|---|---|---|---|
| | | | | *Development* | | | | | |
| BERT$_{base}$ | 92.3 | 84.6 | 91.5 | 91.3 | 60.3 | 89.9 | 85.1 | 70.8 | 83.23 |
| +*SCAL*† [17] | 92.8 | 84.1 | 90.9 | 91.4 | 61.7 | - | - | 69.7 | - |
| +*SuperT*† [48] | 93.4 | 84.5 | 91.3 | 91.3 | 58.8 | 89.8 | 87.5 | 72.5 | 83.64 |
| +*R-Drop*† [18] | 93.0 | 85.5 | 92.0 | 91.4 | 62.6 | 89.6 | 87.3 | 71.1 | 84.06 |
| +AD-DROP | 93.9 | 85.1 | 92.3 | 91.8 | 64.6 | 90.4 | 88.5 | 75.1 | **85.21** |
| RoBERTa$_{base}$ | 95.3 | 87.6 | 92.9 | 91.9 | 64.8 | 90.9 | 90.7 | 79.4 | 86.69 |
| +*R-Drop* [18] | 95.2 | 87.8 | 93.2 | 91.7 | 64.7 | 91.2 | 90.5 | 80.5 | 86.85 |
| +*HiddenCut*† [15] | 95.8 | 88.2 | 93.7 | 92.0 | 66.2 | 91.3 | 92.0 | 83.4 | 87.83 |
| +AD-DROP | 95.8 | 88.0 | 93.5 | 92.0 | 66.8 | 91.4 | 92.2 | 84.1 | **87.98** |
| | | | | *Test* | | | | | |
| BERT$_{base}$ | 93.6 | 84.7 | 90.4 | 89.3 | 52.8 | 85.6 | 81.4 | 68.4 | 80.78 |
| +AD-DROP | 94.3 | 85.2 | 91.6 | 89.4 | 53.3 | 86.6 | 84.1 | 68.7 | **81.65** |
| RoBERTa$_{base}$ | 94.8 | 87.5 | 92.8 | 89.6 | 58.3 | 88.7 | 86.3 | 75.1 | 84.14 |
| +AD-DROP | 95.9 | 87.6 | 93.4 | 89.5 | 58.5 | 89.3 | 87.9 | 76.0 | **84.76** |

## 3.3 Overall Results

We report the overall results of the fine-tuned models in Table 1. We first compare AD-DROP with existing regularization methods on the development sets, including the original fine-tuning, SCAL [17], SuperT [48], R-Drop [18], and HiddenCut [15]. We observe that AD-DROP surpasses the baselines on most of the datasets. Specifically, AD-DROP yields an average improvement of 1.98 and 1.29 points on BERT$_{base}$ and RoBERTa$_{base}$, respectively. We then discuss the performance of AD-DROP on the test sets. Results in Table 1 show that AD-DROP achieves consistent improvement, boosting the average scores of BERT$_{base}$ and RoBERTa$_{base}$ by 0.87 and 0.62, respectively. Besides, compared with large datasets, AD-DROP achieves more gains on small datasets, which are more likely to cause overfitting, illustrating that AD-DROP is more suitable for small data scenarios.

## 3.4 Ablation Study

We conduct ablation experiments on four small datasets to investigate the impact of different components. Due to the limited number of submissions imposed by the GLUE server for evaluation, the results here are reported on the development sets.

**Attribution methods** AD-DROP can be implemented with different attribution methods to generate the mask matrix in Eq. (1), such as integrated gradient attribution (IGA) introduced Eq. (3), attention weights for attribution (AA), and randomly generating the discard region (RD) in Eq. (6). We replace the gradient attribution (GA) in Eq. (5)-(6) with these methods. From Table 2, we can make three observations. First, AD-DROP with gradient-based attribution methods (GA and IGA)

Table 2: Results of ablation studies, in which *r/w* means "replace with" and *w/o* means "without".

| Methods | CoLA | STS-B | MRPC | RTE |
|---|---|---|---|---|
| BERT$_{base}$ | 60.3 | 89.9 | 85.1 | 70.8 |
| +AD-DROP (GA) | **64.6** | 90.4 | **88.5** | **75.1** |
| *r/w* IGA | 63.8 | **90.7** | **88.5** | 74.4 |
| *r/w* AA | 63.6 | 90.0 | 88.0 | 74.7 |
| *r/w* RD | 62.1 | 90.2 | 87.8 | 74.7 |
| *r/w* gold labels | 63.2 | - | 88.0 | 74.4 |
| *w/o* cross-tuning | 62.1 | 90.4 | 87.3 | 71.5 |
| RoBERTa$_{base}$ | 64.8 | 90.9 | 90.7 | 79.4 |
| +AD-DROP (GA) | 66.8 | 91.4 | **92.2** | **84.1** |
| *r/w* IGA | **68.1** | **91.6** | 91.4 | 82.7 |
| *r/w* AA | 66.3 | 91.5 | 91.2 | 82.3 |
| *r/w* RD | 66.5 | 91.5 | **92.2** | 82.0 |
| *r/w* gold labels | 66.4 | - | 91.2 | 82.0 |
| *w/o* cross-tuning | 67.3 | 91.3 | 90.4 | 80.5 |

surpasses that with the other methods (AA or RD) on most of the datasets, illustrating that gradient-based methods are better at finding features that are likely to cause overfitting. Second, IGA outperforms GA in some cases. Although IGA provides better theoretical justification than GA for attribution, it requires prohibitively more computational cost than GA (see Section 4.7 for efficiency analysis), making GA a more desirable choice for AD-DROP. Third, AD-DROP improves the original

BERT$_{base}$ and RoBERTa$_{base}$ with any of the masking strategies, demonstrating the robustness of AD-DROP to overfitting when fine-tuning these models.

**Pseudo labels *vs* gold labels**  In Section 2.3, we discuss the motivation of using pseudo labels for attribution in AD-DROP. To verify the reasonability, we conduct an experiment with gold labels for attribution. As the results show in Table 2, using gold labels for attribution deteriorates the performance, illustrating that AD-DROP with pseudo labels for attribution is preferable.

**Cross-tuning**  To verify the effectiveness of the cross-tuning strategy, we ablate it and apply only AD-DROP in all training epochs. As shown in Table 2, removing cross-tuning causes noticeable performance degradation on most of the datasets. This can be explained by the intuition that AD-DROP without cross-tuning tends to discard high-attribution positions excessively and make the model difficult to converge normally. To vividly demonstrate the effect of AD-DROP with or without cross-tuning, we visualize the distributions of the performance on the RTE[4] development set when enumerating the parameters $p$ and $q$ in the range of [0.1, 0.9]. The results are plotted in Figure 4, where each blue/orange point denotes the accuracy with a pair of $p$ and $q$ values. We observe from the figure that AD-DROP without cross-tuning cannot be trained properly under some parameter settings. However, it works well for most parameter settings when cross-tuning is applied, demonstrating that cross-tuning is vital for improving training stability.

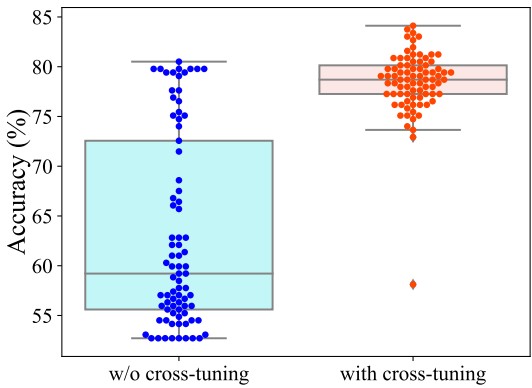

Figure 4: Results of AD-DROP with and without cross-tuning when enumerating $p$ and $q$ in [0.1, 0.9]. RoBERTa is chosen as the base model. Results show that "with cross-tuning" leads to much lower variance and higher performance.

## 4 Analysis

In this section, we further conduct several experiments for more thorough analysis.

### 4.1 Repeated Experiments

To reduce the influence of randomness, we conduct repeated experiments on four small datasets (i.e., CoLA, STS-B, MRPC and RTE). We repeat the training of each model with five random seeds and report the average score and standard deviation on the development sets. From Table 3, we observe that AD-DROP outperforms the original fine-tuning on all

Table 3: Results of repeated experiments. Each score is the average of five runs with a standard deviation.

| Methods | CoLA | STS-B | MRPC | RTE |
|---|---|---|---|---|
| BERT$_{base}$ | $61.8_{\pm 1.9}$ | $89.4_{\pm 0.5}$ | $85.2_{\pm 1.3}$ | $71.2_{\pm 1.2}$ |
| +AD-DROP | $\mathbf{63.4}_{\pm 0.4}$ | $\mathbf{90.1}_{\pm 0.5}$ | $\mathbf{87.4}_{\pm 0.9}$ | $\mathbf{73.9}_{\pm 1.1}$ |
| RoBERTa$_{base}$ | $64.3_{\pm 0.9}$ | $91.0_{\pm 0.2}$ | $89.8_{\pm 0.8}$ | $79.1_{\pm 1.7}$ |
| +AD-DROP | $\mathbf{66.4}_{\pm 0.9}$ | $\mathbf{91.2}_{\pm 0.1}$ | $\mathbf{91.3}_{\pm 0.7}$ | $\mathbf{82.5}_{\pm 0.9}$ |

the datasets. In addition, AD-DROP results in lower standard deviations on most of the datasets, showing that AD-DROP is more robust in fine-tuning PrLMs than the original approach.

### 4.2 Effect of Data Size

To study the impact of data size, we compare AD-DROP with the original fine-tuning (FT) approach on QNLI and QQP,[5] two relatively large datasets, and report their performance when the number of training samples changes. RoBERTa is chosen as the base model. Figure 5 shows that AD-DROP

---

[4]Results on the other datasets are shown in Appendix D.1.

[5]Results on QQP are shown in Appendix D.2.

outperforms FT consistently on QNLI. Moreover, AD-DROP improves the efficiency of data use as training AD-DROP with 60% training samples produces comparable performance to FT with full data.

### 4.3 Hyperparameter Sensitivity

AD-DROP involves two hyperparameters $p$ and $q$ to control the number of discarded attention positions. To investigate the sensitivity of AD-DROP to them, we show the results of different $p$ and $q$ combinations on CoLA and RTE in Figure 6, in which we apply `MaxAbsScaler`[6] to project the difference between the results of AD-DROP and FT into the interval of $[-1.0, 1.0]$. We observe that BERT with AD-DROP is not hyperparameter-sensitive as it outperforms the baseline under most settings. In

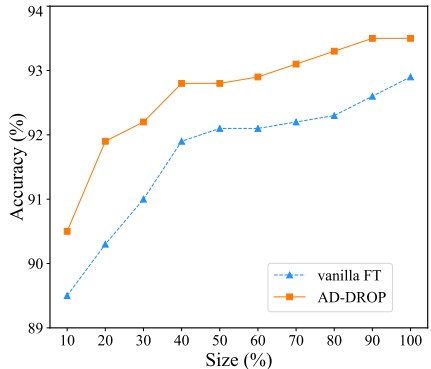

Figure 5: Results of AD-DROP and FT as the number of training samples changes.

contrast, RoBERTa with AD-DROP is more sensitive and requires a careful search for optimal hyperparameter settings. The possible reason is that RoBERTa is pre-trained with more data and more effective tasks than BERT, making it less prone to overfitting than BERT.

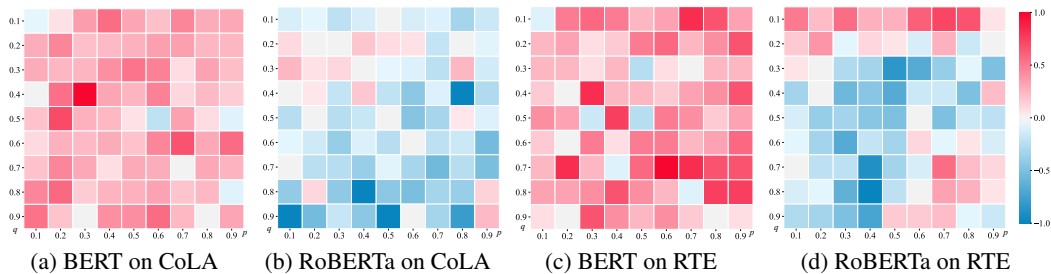

| (a) BERT on CoLA | (b) RoBERTa on CoLA | (c) BERT on RTE | (d) RoBERTa on RTE |

Figure 6: Results of sensitivity study on CoLA and RTE. Rows correspond to $p$ and columns refer to $q$. Blue blocks indicate the results of AD-DROP below the baseline (FT), and red blocks mean the results of AD-DROP above the baseline. Darker colors mean greater gaps with the baseline.

### 4.4 Larger Model Size

To verify the scalability of AD-DROP for a larger model size, we evaluate AD-DROP with RoBERTa$_{\text{large}}$ on the RTE and MRPC datasets. Table 4 shows the average scores and standard deviations of five random seeds. There are two main observations. First, AD-DROP achieves consistent improvements over the larger RoBERTa model, il-

Table 4: Testing AD-DROP on a larger model.

| Methods | MRPC | RTE |
|---|---|---|
| RoBERTa$_{\text{large}}$ | $90.83_{\pm 0.75}$ | $85.99_{\pm 0.86}$ |
| +AD-DROP | $\mathbf{91.62}_{\pm \mathbf{0.53}}$ | $\mathbf{88.01}_{\pm \mathbf{0.48}}$ |

lustrating that AD-DROP is scalable to large models. Second, compared with RoBERTa$_{\text{base}}$ on RTE in Table 3, the larger model significantly reduces the deviation (from 1.7 to 0.86), suggesting that a larger model size indeed helps to improve the stability. AD-DROP further improves the performance and reduces the deviation.

### 4.5 Few-shot Scenario

In this subsection, we test the performance of AD-DROP under few-shot scenarios. Specifically, we carry out 16-, 64-, and 256-shot experiments on SST-2 and CoLA with RoBERTa$_{\text{base}}$ as the base model and the baseline. We report the average scores and standard deviations of five random seeds in Table 5. We observe that RoBERTa with AD-DROP consistently outperforms the original fine-tuning approach. Besides, AD-DROP tends to bring more benefits when fewer samples are available.

---

[6]https://scikit-learn.org/stable/modules/generated/sklearn.preprocessing.MaxAbsScaler.html

Table 5: Testing AD-DROP in few-shot settings. RoBERTa with AD-DROP achieves higher performance and lower deviations than that with the original fine-tuning approach.

| Methods | SST-2 | | | CoLA | | |
| | 16-shot | 64-shot | 256-shot | 16-shot | 64-shot | 256-shot |
|---|---|---|---|---|---|---|
| RoBERTa$_{base}$ | $74.50_{\pm 3.03}$ | $89.06_{\pm 0.83}$ | $91.44_{\pm 0.17}$ | $23.18_{\pm 6.38}$ | $39.70_{\pm 4.68}$ | $51.11_{\pm 1.64}$ |
| +AD-DROP | $\mathbf{80.16_{\pm 1.51}}$ | $\mathbf{91.61_{\pm 0.52}}$ | $\mathbf{92.61_{\pm 0.13}}$ | $\mathbf{26.70_{\pm 4.96}}$ | $\mathbf{46.41_{\pm 1.98}}$ | $\mathbf{52.47_{\pm 1.16}}$ |

## 4.6 Out-of-Distribution Generalization

To further demonstrate AD-DROP is beneficial to reducing overfitting, we test AD-DROP with RoBERTa$_{base}$ on two out-of-distribution (OOD) datasets, i.e., HANS and PAWS-X. For HANS, we use the checkpoints trained on MNLI and test their performance on the validation set (the test set is not supplied). For PAWS-X, we use the checkpoints

Table 6: Testing AD-DROP on OOD datasets.

| Methods | HANS | PAWS-X |
|---|---|---|
| RoBERTa$_{base}$ | 69.83 | 47.90 |
| +AD-DROP | **70.49** | **51.25** |

trained on QQP and examine its performance on the test set. The evaluation metric is accuracy. From Table 6, we can see that RoBERTa with AD-DROP achieves better generalization, where AD-DROP boosts the performance by 0.66 on HANS and 3.35 on PAWS-X, illustrating that the model trained with AD-DROP generalizes better to OOD data.

## 4.7 Computational Efficiency

To analyze the computational efficiency, we quantitatively study the computational cost of AD-DROP with different dropping strategies (GA, IGA, AA, and RD) relative to the original fine-tuning on CoLA, STS-B, MRPC, and RTE. BERT is chosen as the base model for this experiment. As shown in Table 7, although IGA achieves more favorable performance on one of the datasets, it requires higher computational costs than its counterparts, especially when applied in all the layers. In contrast, AD-DROP with GA is more competitive in both performance and computational cost.

Table 7: Results of performance and computational cost of AD-DROP with different masking strategies (GA, IGA, AA, and RD) relative to the original fine-tuning. The symbol ‡ means AD-DROP is only applied in the first layer. BERT is chosen as the base model.

| Methods | CoLA | | STS-B$^{\ddagger}$ | | MRPC | | RTE | |
| | Mcc | Time | Pcc | Time | Acc | Time | Acc | Time |
|---|---|---|---|---|---|---|---|---|
| RD | +1.8 | ×1.42 | +0.3 | ×1.38 | +2.7 | ×1.31 | +3.9 | ×1.42 |
| AA | +3.3 | ×1.42 | +0.1 | ×1.48 | +2.9 | ×1.94 | +3.9 | ×1.58 |
| GA | +4.3 | ×3.58 | +0.5 | ×1.95 | +3.4 | ×4.13 | +4.3 | ×4.50 |
| IGA | +3.5 | ×99.61 | +0.8 | ×15.00 | +3.4 | ×110.12 | +3.6 | ×125.67 |

## 5 Related Work

**Dropout** Dropout is a widely used regularizer to mitigate overfitting when training deep neural networks. Vanilla dropout [19] randomly selects neurons with a predefined probability and sets their values to zeros during training. By doing so, the neurons cannot co-adapt and the trained networks can lead to better generalization. In recent years, many variants of dropout have emerged. One body of research aims to adopt different strategies to drop units in neural networks. For example, DropConnect [49] randomly selects connections between neurons to discard. DropBlock [21] defines a structured dropout that randomly drops the units in a specific contiguous region of a feature map. AutoDropout [22] aims to improve the dropout pattern of DropBlock by introducing an automatic method to design dropout structures. HiddenCut [15] drops contiguous spans within the hidden space, in which the attention weights are utilized to select the dropped spans strategically.

Another body of research devotes to addressing the inconsistency between training and inference when dropout is applied. For instance, mixout [11] randomly replaces selected parameters with original pre-trained weights rather than setting them to zeros. CHILD-TUNING [13] selects a child network and masks out the gradients of the non-child network during the backward step, only updating weights in the child network. R-Drop [18] performs dropout twice in the forward steps to produce two sub-models and then applies KL-divergence for their output distributions, forcing the two sub-models to be consistent with each other. However, most of these methods follow the random sampling strategy of dropout and pay little attention to the different importance of self-attention positions in PrLMs.

**Attribution** Numerous studies have been undertaken to interpret the behaviors of deep neural networks (DNNs). As a theory for interpretability, attribution aims to evaluate the impact of input features on predictions [24]. Generally, attribution methods can be divided into perturbation-based [50–52], gradient-related [25, 53, 26], and attention-based [54–56] methods. We focus on reviewing the gradient-related methods as they are more relevant to our work. Specifically, earlier works [25, 57, 58] try to explain model decisions via gradients since gradients indicate the direction and rate that changes the loss the fastest. However, Sundararajan et al. [26] point out that gradient attribution violates the sensitivity axiom in some cases that the gradients will be zero for the function in saturated areas, and propose integrated gradient as a theoretically more sound attribution method.

Other efforts have been devoted to revealing the behavior patterns of PrLMs. Kovaleva et al. [8] and Clark et al. [9] use attention weights for attribution to investigate what specific knowledge BERT [1] learns. Jain and Wallace [27] and Brunner et al. [59] investigate the identifiability of attention weights and conclude that attention weights are not a faithful explanation for model predictions. Hao et al. [23] apply integrated gradient [26] as a saliency measure for self-attention attribution in BERT, and use the attribution result to interpret information interactions inside Transformers. Similarly, Lu et al. [28] develop influence patterns based on integrated gradient to explain information flow in BERT. Unlike these works, we aim to examine the effect of dropout on self-attention through self-attention attribution and to reduce overfitting when fine-tuning PrLMs.

## 6    Conclusion

We propose a novel dropout regularizer, AD-DROP, to mitigate overfitting when fine-tuning PrLMs on downstream tasks. Unlike previous dropout-based methods that generally adopt the random sampling strategy to discard units, AD-DROP draws inspiration from self-attention attribution which reveals that attention positions are not equally important in reducing overfitting and that dropping inappropriate positions may exacerbate the problem. Therefore, AD-DROP focuses on discarding high-attribution attention positions to prevent the model from relying heavily on these positions to make predictions. Besides, we propose a cross-tuning strategy that performs the original fine-tuning and our AD-DROP alternately to stabilize the fine-tuning process. Extensive experiments and analysis on the GLUE benchmark demonstrate the effectiveness of AD-DROP. Although originally proposed and evaluated based on self-attention attribution, AD-DROP can be potentially extended to other neural network units as vanilla dropout, which deserves further research efforts.

## Acknowledgments

We appreciate the anonymous reviewers for their valuable comments. We thank Feifan Yang, Zihong Liang, and Hong Ding for their early discussions. This work was supported by the National Natural Science Foundation of China (No. 62176270), the Program for Guangdong Introducing Innovative and Entrepreneurial Teams (No. 2017ZT07X355), and the Foundation of Key Laboratory of Machine Intelligence and Advanced Computing of the Ministry of Education.

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
