# A  Appendix: AD-DROP for Token-Level Tasks

## A.1  Attribution Matrix

Note that AD-DROP is naturally suitable for classification tasks since we can obtain one single attribution matrix with respect to the only logit output for each attention map. For token-level tasks (e.g., NER and text generation), as we have several logit outputs to produce the corresponding attribution matrices for each attention map, applying AD-DROP has the challenge of how to fuse these attribution matrices. We provide a simple alternative to calculate the attribution matrix as:

$$\widetilde{\mathbf{B}}_h = -\frac{\partial \mathcal{L}}{\partial \mathbf{A}_h}, \tag{1}$$

where $\mathcal{L}$ is the pseudo loss in terms of the pseudo labels. Given a sequence $x$ with $n$ input tokens, we represent each pseudo label as a one-hot vector of $C$ elements and compute $\mathcal{L}$ as:

$$\mathcal{L} = \sum_{i=1}^{n} \mathcal{L}_i = -\sum_{i=1}^{n}\sum_{c=1}^{C} y_{i,c}\log P_F(c|x,i) = -\sum_{i=1}^{n} y_{i,\tilde{c}}\log P_F(\tilde{c}|x,i), \tag{2}$$

where $y_{i,c}$ is the $c$-th element in the one-hot vector for token $i$, $P_F(c|x,i)$ is the softmax output of class $c$ for token $i$, and $\tilde{c}$ is the pseudo label. Then, Eq. (1) can be updated as:

$$\widetilde{\mathbf{B}}_h = -\frac{\partial \mathcal{L}}{\partial \mathbf{A}_h} = -\sum_{i=1}^{n}\frac{\partial \mathcal{L}_i}{\partial F_{i,\tilde{c}}(\mathbf{A})} \cdot \frac{\partial F_{i,\tilde{c}}(\mathbf{A})}{\partial \mathbf{A}_h} = \sum_{i=1}^{n}(y_{i,\tilde{c}} - P_F(\tilde{c}|x,i))\,\mathbf{B}_{i,h}. \tag{3}$$

Therefore, we can use Eq. (3) to compute a single attribution matrix for each attention map when applying AD-DROP in token-level tasks. Besides, as regression tasks (e.g., STS-B) cannot infer pseudo labels, we directly use the actual loss instead.

## A.2  Token-Level Experiments

We conduct additional experiments of AD-DROP on NER (CoNLL-2003) and Machine Translation (WMT 2016) tasks.[1] The results on the test sets are reported in Table 1 and Table 2. Moreover, to verify that AD-DROP can be adapted to other pre-trained models, for CoNLL-2003 NER, we choose ELECTRA as the base model. For WMT 2016, OPUS-MT is chosen. The results show that AD-DROP consistently improves the baselines on both NER and Machine Translation tasks.

Table 1: Test results of AD-DROP on the CoNLL-2003 NER dataset.

| Methods | Accuracy | F1 |
|---|---|---|
| ELECTRA$_{\text{base}}$ | 97.83 | 91.23 |
| +AD-DROP | **97.95** | **91.77** |

Table 2: Test results of AD-DROP on Translation datasets. The evaluation metric is BLEU.

| Methods | EN-RO | TR-EN |
|---|---|---|
| OPUS-MT | 26.11 | 23.88 |
| +AD-DROP | **26.43** | **23.96** |

# B  Appendix: More Prior Experiments

Our observations show that dropping low-attribution positions makes the model fit the training data rapidly, while dropping high-attribution positions reduces the fitting speed. To further probe the effect of dropping low- or high-attention positions, we fine-tune a RoBERTa on the training set and evaluate its performance on the development set by applying the two dropping strategies. The results on MRPC, SST-2, and QNLI are plotted in Figure 1. Similar phenomena can be observed that the model rapidly fits the data while dropping only a small proportion of low-attribution positions. As the dropping rate increases, the accuracy remains stable until discarding too many low-attribution positions. When dropping high-attribution positions, we observe an opposite trend that the performance deteriorates sharply. These results further confirm the observations in Section 2.2 that attention positions should not be treated equally important in dropout.

---

[1]We follow the official colab implementation (`https://huggingface.co/transformers/v4.7.0/notebooks.html`) for the two tasks.

Note that we only drop positions in the first layer of RoBERTa for the above experiments to exclude the impact of different layers. We also conduct experiments in the other layers on SST-2, and the overall results are shown in Figure 2. We note that similar results are obtained in the first few layers, while the trend becomes less obvious as the number of layers increases. It could be caused by the over-smoothing issue that the representations of all tokens are similar in the last few layers.

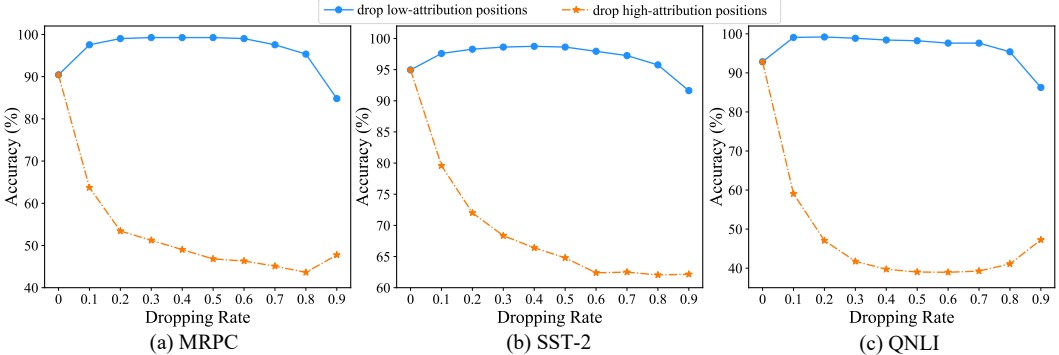

Figure 1: Performance of fine-tuned RoBERTa on development sets, where two dropping strategies (i.e., drop low-/high-attribution positions) are applied. Gold labels are used for the attribution.

# C   Appendix: Experimental Details

## C.1   Details of Datasets

The details of the used datasets are introduced as follows. (1) Stanford Sentiment Treebank (**SST-2**) is a sentence sentiment prediction task. (2) Multi-Genre Natural Language Inference (**MNLI**) is a pairwise sentence classification task that aims to predict whether the relationship between two sentences is entailment, contradiction, or neutral. (3) Question Natural Language Inference (**QNLI**) is a binary sentence classification task that aims to predict whether the sentence in a question-sentence pair contains the correct answer to the question. (4) Quora Question Pairs (**QQP**) is a binary pairwise sentence classification task that aims to predict whether two questions are semantically equivalent. (5) The Corpus of Linguistic Acceptability (**CoLA**) aims to predict whether a single English sentence conforms to linguistics. (6) The goal of Semantic Textual Similarity Benchmark (**STS-B**) is to pre-

Table 3: Statistics of the used datasets.

| Dataset | Train | Dev | Test |
|---|---|---|---|
| SST-2 | 67349 | 872 | 1821 |
| MNLI | 392702 | 9815 | 9796 |
| QNLI | 104743 | 5463 | 5463 |
| QQP | 363846 | 40430 | 390965 |
| CoLA | 8551 | 1043 | 1063 |
| STS-B | 5749 | 1500 | 1378 |
| MRPC | 3668 | 408 | 1725 |
| RTE | 2490 | 277 | 3000 |
| CoNLL-2003 | 14041 | 3250 | 3453 |
| EN-RO | 610320 | 1999 | 1999 |
| TR-EN | 205756 | 1001 | 3000 |
| HANS | 30000 | 30000 | - |
| PAWS-X | 49401 | 2000 | 2000 |

dict how two given sentences are semantically similar. (7) Microsoft Research Paraphrase Corpus (**MRPC**) aims to predict if two sentences are semantically equivalent. (8) Recognizing Textual Entailment (**RTE**) is similar to MNLI but has binary labels. (9) **CoNLL-2003** is to recognize the named entities in a sentence, which contains four types of named entities. (10) WMT 2016 is a multilingual translation database. In this study, we choose **English-Romanian (EN-RO)** and **Turkish-English (TR-EN)** for the experiment. (11) Heuristic Analysis for NLI Systems (**HANS**) aims to evaluate whether NLI models adopt syntactic heuristics. (12) **PAWS-X** is a cross-lingual adversarial dataset for paraphrase identification. HANS and PAWS-X are typically used for the OOD generalization test. The statistics of these datasets are shown in Table 3.

## C.2   Hyperparameter Settings

Table 4 presents the final hyperparameter settings of AD-DROP for BERT/RoBERTa$_{base}$. The setting with only one value means the parameter is shared by BERT and RoBERTa.

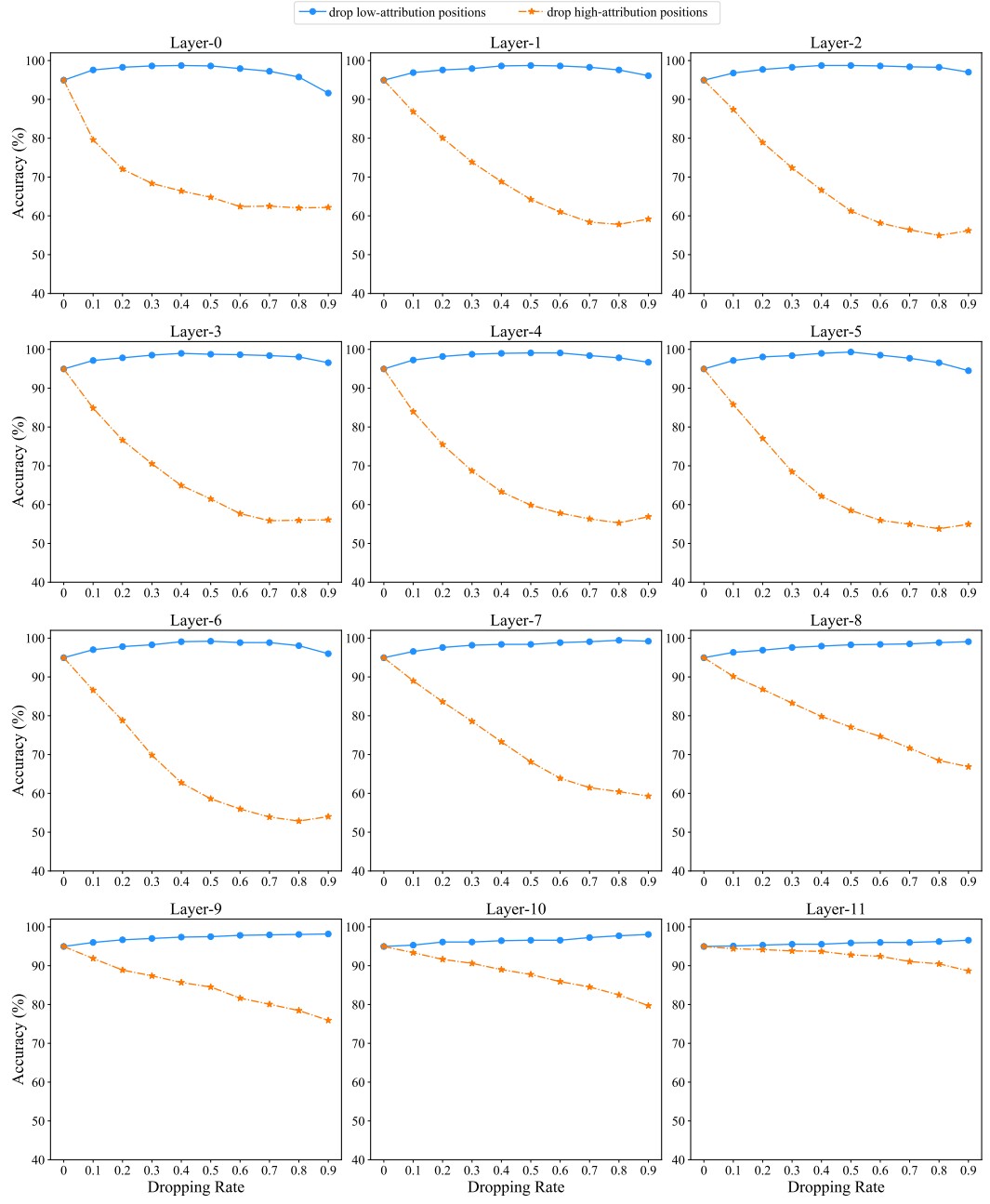

Figure 2: Results of dropping self-attention positions in different layers of RoBERTa on SST-2.

Table 4: Hyperparameter settings of AD-DROP for BERT and RoBERTa.

| Dataset | Learning rate | Batch size | Length | $p$ | $q$ |
|---------|---------------|------------|--------|-----|-----|
| SST-2 | 1e-5 | 16/64 | 120 | 0.6/0.3 | 0.8/0.7 |
| MNLI | 1e-5 | 16/32 | 128 | 0.5/0.4 | 0.9/0.2 |
| QNLI | 1e-5 | 16 | 128 | 0.8 | 0.8/0.4 |
| QQP | 1e-5 | 16 | 120 | 0.2/0.7 | 0.7/0.9 |
| CoLA | 1e-5/2e-5 | 16 | 47 | 0.3/0.8 | 0.4/0.3 |
| STS-B | 1e-5/2e-5 | 16 | 100 | 0.9/0.1 | 0.7/0.5 |
| MRPC | 1e-5/2e-5 | 16 | 100 | 0.5/0.8 | 0.8/0.3 |
| RTE | 1e-5 | 16 | 128 | 0.6/0.7 | 0.7/0.1 |

# D   Appendix: More Experimental Results

## D.1   Ablation of Cross-Tuning

We further report the results of removing cross-tuning in AD-DROP when enumerating $p$ and $q$ in the range of [0.1, 0.9] on the CoLA and MRPC datasets. We observe consistent performance degradation in Figure 3 after removing the cross-tuning strategy from AD-DROP.

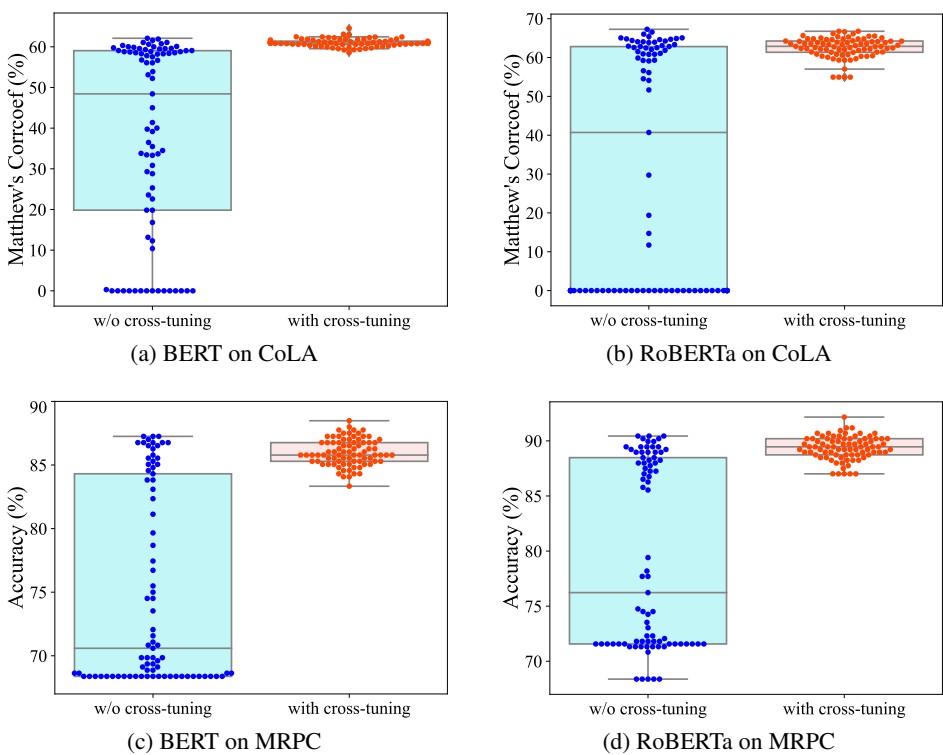

(a) BERT on CoLA    (b) RoBERTa on CoLA

(c) BERT on MRPC    (d) RoBERTa on MRPC

Figure 3: Results of AD-DROP with and without cross-tuning when enumerating $p$ and $q$ in the range of [0.1, 0.9] on the CoLA and MRPC datasets.

## D.2   Effect of Data Size on QQP

Figure 4 shows a comparison between AD-DROP and the original fine-tuning (FT) as the size of training examples changes. We observe from the figure that AD-DROP performs consistently better than original FT with different sizes of training data.

# E   Appendix: Limitations

We discuss potential limitations of AD-DROP as follows. First, as reported in Section 4.7, training with AD-DROP requires more computational cost than the original fine-tuning approach especially when integrated gradient is applied for attribution in all the attention heads. Therefore, we propose to use gradient for attribution in AD-DROP as it achieves competitive performance with acceptable computational cost. Second, AD-DROP introduces additional hyperparameters ($p$ and $q$) and requires more effort to search for the best hyperparameters.

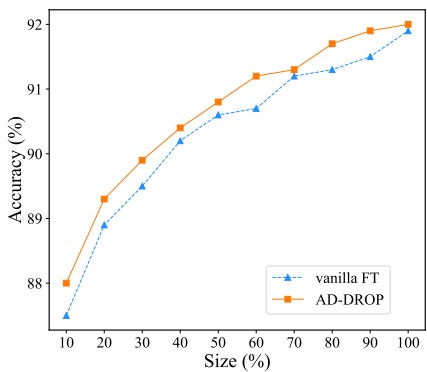

Figure 4: Results of comparison between AD-DROP and original FT as the size of training data changes on QQP.