# OpenReview forum: "AD-DROP: Attribution-Driven Dropout for Robust Language Model Fine-Tuning"
_NeurIPS.cc/2022/Conference — NeurIPS 2022 Accept_

### Official Review · Reviewer_VyWF · 2022-06-28

**Rating:** 5
**Confidence:** 3
**Soundness:** 3 good
**Presentation:** 3 good
**Contribution:** 2 fair

**Summary:**

This work proposes AD-DROP to alleviate overfitting. AD-DROP involves four steps. First, predictions are made through a  forward computation without dropping any attention position. Second, compute the attribution score of each attention position by attribution methods. Third,  randomly sample a set of positions with high attribution scores and generate a masking matrix for each attention map. Finally, the masking matrices are applied to the next forward computation to make predictions for backpropagation.

In experiments, it is shown that AD-DROP is stable, and gives visible improvements over strong regularization baselines.

**Questions:**

Question:
In equation6, what is n?
I don't quite get the intuition behind cross-tuning, why not just use a smaller dropout probability?
In Table 1, why is the baselines in bert and roberta different?

**Limitations:**

Yes.

**Strengths And Weaknesses:**

Strength:
The method is novel in that it involves attribution methods. Instead of pure random.
In experiments it outperform some strong baselines.

Weakness:
Experiments maybe not complete.(refer to my questions)
Since AD-Drop could prevent overfitting, I'd like to see how it performs in few-shot learning. (Sec4.2 is helpful, but I think it's better to see more datasets)
The attribution method would cost some computation.

---

> ### Author Response · Authors · 2022-08-02
> **Response to Reviewer VyWF**
>
> We appreciate your valuable comments and try to address your concerns as follows.
> * **Q1: Since AD-DROP could prevent overfitting, I'd like to see how it performs in few-shot learning. (Sec4.2 is helpful, but I think it's better to see more datasets)**
>     >**A1:** Good suggestion. We carry 16, 64, and 256-shot experiments on SST-2 and CoLA with RoBERTa-base as the base model and the baseline. As shown in the table below, we report the average scores and standard deviations of five random seeds. We observe that RoBERTa with AD-DROP consistently outperforms the original finetuning approach. Besides, AD-DROP tends to bring more benefits when fewer samples are available.
>
>     > **Table: Testing AD-DROP on few-shot settings.**
>     | Methods      | SST-2(16-shot) | SST-2(64-shot) | SST-2(256-shot) | CoLA(16-shot) | CoLA(64-shot)  | CoLA(256-shot) |
>     |:------------:|:--------------:|:--------------:|:---------------:|:-------------:|:--------------:|:--------------:|
>     | RoBERTa-base | 74.50$\pm$3.03 | 89.06$\pm$0.83 | 91.44$\pm$0.17  | 23.18$\pm$6.38 | 39.70$\pm$4.68 | 51.11$\pm$1.64 |
>     | +AD-DROP     | **80.16$\pm$1.51** | **91.61$\pm$0.52** | **92.61$\pm$0.13**  | **26.7$\pm$4.96**  | **46.41$\pm$1.98** | **52.47$\pm$1.16** |
>
> * **Q2: The attribution method would cost some computation.**
>     > **A2:** Like many other regularization techniques, the attribution method inevitably causes additional costs, but it still has a vast space to optimize. For example, Figure 8 illustrates that layers are not of the same importance. Hence, we can take layer importance into account when performing AD-DROP, which helps reduce the costs considerably.
>
> * **Q3: In equation6, what is $n$?**
>     > **A3:** As defined in Line 77, $n$ in Equation 6 means the length of an input sequence.
>
> * **Q4: I don't quite get the intuition behind cross-tuning, why not just use a smaller dropout probability?**
>     > **A4:** The intuition of cross-tuning is to alternate finetuning and AD-DROP to avoid dropping high attribution positions excessively. Another intuitive idea is to set a smaller dropout probability for AD-DROP. However, we found it increases the difficulty of tuning hyperparameters $p$ and $q$ and limits the adjustability of AD-DROP. Hence, cross-tuning appears to be a better trade-off between dropping too many positions and stable training.
>
> * **Q5: In Table 1, why is the baselines in BERT and RoBERTa different?**
>     > **A5:** The results of baselines in Table 1 are directly taken from the original papers, while most of them are reported either on BERT or RoBERTa, which causes the difference.

---

> > ### Comment · Reviewer_VyWF · 2022-08-03
> > **Thanks**
> >
> > These are helpful, especially the fewshot experiment, thanks!

---

### Official Review · Reviewer_PC5L · 2022-07-09

**Rating:** 4
**Confidence:** 4
**Soundness:** 2 fair
**Presentation:** 3 good
**Contribution:** 2 fair

**Summary:**

This paper presents a new fine-tuning strategy called AD-DROP to prevent overfitting for PLM on language understanding tasks based on the attribution scores on attention positions. The idea is intuitive that dropping attention positions with low attribution scores increases the chance of overfitting. Based on this observation, this paper proposes to randomly discard some of the attention distribution with high attribution scores to make the model exploit more information from low attribution positions thus it is less likely to overfit. The empirical results on NLU tasks demonstrate the success of the proposed strategy, especially on small datasets.

**Questions:**

See Weakness.

**Limitations:**

Yes

**Strengths And Weaknesses:**

**Strengths**
- This paper is well-written and easy to understand.
- The proposed method is simple and easy to apply.
- The empirical results on NLU tasks look promising compared to full fine-tuning approaches.


**Weakness**
- Limited tasks and analysis. Experiments are only conducted on GLUE, which leaves questions on the applicability of the proposed approach. Other than the aspect indicated by the authors that the AD-DROP could potentially be applied to different units in the model, there are still multiple dimensions that could have been considered to make the evaluation of the approach more thorough, convincing, and insightful. For example, 1) only Masked PLMs are considered; what happens if the base model is an autoregressive LM? 2) Other than NLU tasks, can it be applied to generation tasks? 3) How the approach scales with model sizes? In terms of overfitting, both limited data and large model size could be the source and this should also be investigated.
- The motivation for including the hyper-parameter $q$ is unclear. I understood it is used to control the strength of enforcing the masking to the softmax, but isn't this sort of covered by the $p$? It seems to me that a random sampling based on uniform distribution to decide the positions (in addition to Eq 6) could do the job. Including $q$ seems to require much more hyper-parameter tuning. Besides, no description of $q$ is given around Line 145, I had to guess what $q$ is.
- Some confusion about the experiments/analysis
> 1. In Table 1, the improvement of R-Drop with RoBERTa Base seems quite limited compared to the improvement with RoBERTa Large on the original paper. I assume the authors implemented and ran the R-Drop with RoBERTa. However, there is no description of how they chose the hyper-parameter for R-Drop and the search space for that.
>2. Comparing Table 3 to Table 1, it looks like the variance of the improvement is rather notable. For example, on RTE with BERT, the improvement reported in Table 1 is **4.3** while when averaging over multiple rounds, the improvement in Table 3 is **2.7**.  It looks to me that it is more reasonable to report averaging numbers in Table 1 rather than the best possible numbers that are ever achieved?
>3. In Table 4, I may misunderstand something here. It looks to me that it is almost impossible that the computation cost of AD-DROP on STS-B is the same as FT. AD-DROP needs **TWO** forward passes of the model. It requires running a **forward pass** of the model first to obtain the pseudo label as well as the **attribution scores computation**. Lastly, the model will run the forward pass again for back-propagation. On the other hand, FT only requires **ONE** forward pass and back-propagation. Would you clarify how the computational cost is defined and calculated here such that they are the same?

---

> ### Author Response · Authors · 2022-08-02
> **Response to Reviewer PC5L**
>
> We appreciate your valuable comments and try to address your concerns as follows.
> * **Q1: Regarding additional experiments on other PLMs and tasks, and how the approach scales with model sizes.**
>     > **A1:** Many thanks for the insightful comments. We conduct additional experiments on NER and Machine Translation tasks with other PLMs, and the results verify the effectiveness of AD-DROP. Please refer to the **[General Response](https://openreview.net/forum?id=XYDXL9_2P4&noteId=9Idjo23DzV)** for details.
>
>     > Besides, we investigate the impact of model sizes. The table below shows the results of AD-DROP on RoBERTa-large. We observe that AD-DROP achieves consistent improvements over the larger RoBERTa model, illustrating that AD-DROP is scalable to large models.
>
>     > **Table: Testing AD-DROP on RoBERTa-large.**
>     | Methods       | MRPC         | RTE           |
>     |---------------|--------------|---------------|
>     | RoBERTa-large | 90.83$\pm$0.75 | 85.99$\pm$0.86  |
>     | +AD-DROP      | **91.62$\pm$0.53** | **88.01$\pm$0.48**  |
>
> * **Q2: Regarding the motivation for including the hyper-parameter $q$.**
>     > **A2:** We hope to provide some clarification here. $p$ is used to define a **candidate** discard area (e.g., top 30% positions based on the attribution scores), and then we randomly drop certain amount of attention positions, controlled by $q$, within this candidate discard area (e.g., randomly drop 10% attention positions within the top 30%). Thank you for pointing this out, and we will make the description of $q$ more clear in the revision.
>
> * **Q3: Regarding the implementation of R-Drop and the search space.**
>     > **A3:** To finetune RoBERTa-base with R-Drop, we employed the code released by the original paper and replaced the pretrained checkpoint with RoBERTa-base. Following the original paper, the hyper-parameter $\alpha$ was searched among {0.1, 0.5, 1.0}, and the best results were reported. We will clarify this in the revised paper.
>
> * **Q4: Regarding the results on Table 1 and Table 3.**
>     >**A4:** The purpose of Table 1 was to compare AD-DROP with several existing baselines, some of which (e.g., HiddenCut and R-Drop) have not reported repeated results. Since we directly took their results from the original papers, to make a fair comparison, we only reported the best results in Table 1. Thank you for the suggestion and we will revise this table accordingly.
>
> * **Q5: Regarding the computation cost of AD-DROP on STS-B in Table 4.**
>     >**A5:** This is indeed a great observation. We agree with the reviewer that AD-DROP requires two forward pass while FT only needs one. Actually, this is the main reason why AD-DROP requires more computational cost (even with RD or AA) than FT. However, for STS-B, AD-DROP is **only applied to the first layer** instead of all 12 layers (please refer to Q8 in the response to reviewer XaQX on why only applying to the first layer), and thus is able to achieve similar computation cost to FT. The precise time cost of RD and AA are 1.02 and 1.04 respectively, instead of 1.0. We will revise these numbers accordingly in Table 4. Thank you again for the great question!

---

> > ### Comment · Reviewer_PC5L · 2022-08-08
> > **Thank you for the responses**
> >
> > Thank you for the responses and the additional experiment. The responses have addressed my previous concerns. A follow-up question for the MT experiment is how the experiment setup is. I am asking this because the En-Ro results look pretty off to me. For example, a Transformer base model trained on WMT16 En-Ro should give a BLEU score of around $33.xx$.

---

> > > ### Author Response · Authors · 2022-08-08
> > > **Response to Reviewer PC5L**
> > >
> > > Thanks for the question. We conduct the MT experiments following the official colab (https://colab.research.google.com/github/huggingface/notebooks/blob/main/examples/translation.ipynb) with the pretrained OPUS-MT model, and report the BLEU results on the **test set** after five epochs. We found that the BLEU scores of OPUS-MT on EN-RO in the leaderboard (https://paperswithcode.com/sota/sequence-to-sequence-language-modeling-on-1), which are reported on the **validation set**, are consistently around $28$.
> > >
> > > We also realize that the Transformer-based SOTA methods, such as DeLighT, achieve around $34$ in terms of BLEU on WMT2016 EN-RO task (as shown in https://paperswithcode.com/sota/machine-translation-on-wmt2016-english-1). Thank you for pointing this out, and we will supplement additional results of AD-DROP on DeLighT as well in the revised paper.

---

> ### Author Response · Authors · 2022-08-08
> **Look forward to your feedback**
>
> Dear Reviewer PC5L,
>
> Thank you for your valuable suggestions and constructive feedback. We have added the response to your comments. It would be really appreciated if you could let us know for any further questions or comments.
>
> Best regards,
> Authors

---

### Official Review · Reviewer_XaQX · 2022-07-11

**Rating:** 6
**Confidence:** 3
**Soundness:** 2 fair
**Presentation:** 2 fair
**Contribution:** 2 fair

**Summary:**

* The paper presents a novel variant of dropout for self-attention layers in Transformers, by dropping self attention units with high attribution scores. The authors motivate this by showing that dropping low attribution scores leads to faster overfitting, and by dropping high attribution units, the model tends to overfit less.
* The authors also present cross-tuning (alternatively training with and without AD-Drop across different epochs) as a method to counter excessively dropping high attribution scores
* Results presented on GLUE benchmark shows that the AD Drop helps improve performance over vanilla dropout methods
* The authors also present additional ablation experiments, improvements across varying dataset sizes, sensitivity to choices of hyper-parameters and compute efficiency analysis of the proposed method.

[Update]
Updated scores based on author response.

**Questions:**

Questions:
* Why is cross-tuning done at an epoch level ? How do the results change if it were done at a say minibatch level ?
* Given that size of STS-B is smaller than CoLA, why was AD-Drop only applied to the first layer for STS-B (Line 175) ?

**Limitations:**

I think including the limitations section as a part of the main paper, as opposed to the last section of the Appendix, would be more informative in terms of informing the readers of the limitations of the work. Additionally, in it's current form, one of the big limitations that is not addressed is the lack of diversity of fine-tuning tasks: if the proposal of the paper is a new general purpose regularization method, I think not having additional tasks is a big limitation.

**Strengths And Weaknesses:**

Strengths:
* The paper presents a novel regularization method. The idea of leveraging saliency maps to inform which units to drop is quite interesting
* The results for base model training show decent improvements over baseline methods.

Weaknesses
* Given that the paper presents AD Drop as a new regularization method, the results would be more convincing if carried out on a diverse set of tasks. Right now, the results presented are only for classification tasks. Even if this was for encoder only models, showing results on a token level task (eg: SQuAD QnA, NER etc) would help show the generality of the proposed method. Eg: [1] also propose a novel regularization methods, and demonstrate it's applicability across Machine Translation, Summarization, GLUE, Language Modeling and Image Classification.
* While the presented method is quite interesting, the authors don't address _why_ it works. If the hypothesis is that dropping high attribution self-attention units makes the model rely less on spurious features, it would be good to show that the proposed method does better on OOD datasets (eg: HANS for MNLI, PAWS-X for QQP etc).
* It would be good to report mean of 4 / 5 seeds for all the tasks (not just the small tasks). From Table 1, if we account for the mean of 4 seeds for small tasks, then the overall gain from AD-Drop reduces to 84.7 (taking results from Table 3). Thus the gains might be a bit over-stated, and it would be good to account for the variance due to random seeds in the results.
* It is very hard to understand the impact of different hyper-parameter values from Figure 4. Instead, it might be more informative to plot the accuracy as a function of the parameter values for with and without cross-tuning.
* Since that the compute cost can go up by as much as 4.5x, I think it would be informative to compare against other (non-dropout related) strategies for regularization (eg: Sharpness Aware Minimization; see [2]). For [2], the authors show consistent gains at an 2x performance hit (not considering the efficient SAM results). Thus, it would be useful to see how the proposed method compares against that.
* [Minor] The description of candidate discard region is a bit misleading (Line 140). If my understanding is correct, then candidate discard region S_{i,j} = 1implies the self-attention logit is kept, and not discarded.

References

[1] Wu, Lijun, et al. "R-drop: Regularized dropout for neural networks." Advances in Neural Information Processing Systems 34 (2021): 10890-10905.

[2] Bahri, Dara, Hossein Mobahi, and Yi Tay. "Sharpness-aware minimization improves language model generalization." arXiv preprint arXiv:2110.08529 (2021).

---

> ### Author Response · Authors · 2022-08-02
> **Response to Reviewer XaQX**
>
> We appreciate your valuable comments and try to address your concerns as follows.
> * **Q1: Regarding a diverse set of tasks.**
>     > **A1:** Thanks. In response to this suggestion, we conduct additional experiments on NER (CoNLL-2003) and Machine Translation (WMT2016) tasks. The results validate the effectiveness and generalizability of AD-DROP. Please refer to the **[General Response](https://openreview.net/forum?id=XYDXL9_2P4&noteId=9Idjo23DzV)** for details.
>
> * **Q2: Regarding why AD-DROP works and evaluating on OOD datasets.**
>     > **A2:** Good question. Our observation is that discarding a set of high attribution attentions prevents the model from overly relying on high attribution attentions, and our hypothesis is that dropping high attribution self-attention units makes the model rely less on spurious/shortcut features. Prior experiments in Figure 2 demonstrate that dropping high attribution attentions slows the fitting speed, so the model is more likely to seek global optimization during training.
>
>     > Besides, following the suggestion, we test RoBERTa and RoBERTa+AD-DROP on two OOD datasets. For HANS, we use the checkpoints trained on MNLI and test its performance on the validation set (test set is not supplied). For PAWS-X, the checkpoints are trained on QQP, and we examine its performance on the test set. The evaluation metric is accuracy. From the below results we can see that RoBERTa with AD-DROP achieves better generalization, where AD-DROP boosts the performance by 0.66 on HANS and 3.35 on PAWS-X. We will incorporate these results and provide more discussion during revision.
>
>     > **Table: Testing AD-DROP on out-of-distribution datasets.**
>     | Methods      | HANS for MNLI | PAWS-X for QQP  |
>     |--------------|---------------|-----------------|
>     | RoBERTa-base | 69.83         | 47.90           |
>     | +AD-DROP     | **70.49**         | **51.25**           |
>
> * **Q3: Regarding repeated experiments with mean and deviation.**
>     > **A3:** We notice the variances are large on small datasets. Hence, we conduct repeated experiments in Table 3 to exclude the impact of randomness. Thank you for the suggestion, and we will include the repeated results on large datasets in Table 1.
>
> * **Q4: Regarding plotting the accuracy as a function of the parameter values in Figure 4.**
>     > **A4:** Thank you for the suggestion. We will revise it accordingly.
>
> * **Q5: Regarding comparing with other (non-dropout related) strategies for regularization.**
>     > **A5:** Although our costs can be 4.5x, it still has a vast space to optimize. For example, Figure 8 illustrates that layers are not of the same importance. Hence, we can take layer importance into account when performing AD-DROP, which helps reduce the costs considerably. Thanks for pointing this out and providing the reference. We will test AD-DROP on T5 and compare it with SAM.
>
> * **Q6: Regarding the description of candidate discard region (Line 140).**
>     > **A6:** Yes, you are right. The value 1 means the elements are kept. We will make this clear during revision.
>
> * **Q7: Why is cross-tuning done at an epoch level? How do the results change if it were done at a say minibatch level?**
>     > **A7:** In fact, we have tested cross-tuning at the batch level before. But it is not stable at the beginning of training and with poor evaluation performance than epoch level. We believe it is because AD-DROP needs a relatively good model for better attribution, while cross-tuning at the batch level makes the model difficult for attribution as the model only processes limited batch data, especially in the early training stage.
>
> * **Q8: Why was AD-DROP only applied to the first layer for STS-B (Line 175)?**
>     > **A8:** Although smaller than CoLA, STS-B is more stable when finetuning. As shown in Table 3, the standard deviation is less than CoLA (0.5 vs. 1.9 on BERT and 0.2 vs. 0.9 on RoBERTa). Since STS-B is a regression task, we hypothesize that it is less likely to cause overfitting. Actually, we have conducted AD-DROP in all layers on STS-B and found that applying AD-DROP to the first layer can obtain better results on STS-B.

---

> > ### Comment · Reviewer_XaQX · 2022-08-06
> > **Update on scores post author response**
> >
> > In my opinion, the authors have done a commendable job addressing presented concerns and the additional experiments definitely strengthen the paper. I am still not completely satisfied with the conjecture that the authors provide regarding why the method works (i.e "Prior experiments in Figure 2 demonstrate that dropping high attribution attentions slows the fitting speed, so the model is more likely to seek global optimization during training." is fairly hand wavy, and more importantly, why/how does slower fitting speed relate to seeking global optimization is still quite unclear to me). But that said, given that the additional experiments strengthen the claims of the paper, I am updating my scores.

---

### Official Review · Reviewer_XQcd · 2022-07-13

**Rating:** 7
**Confidence:** 4
**Soundness:** 4 excellent
**Presentation:** 4 excellent
**Contribution:** 3 good

**Summary:**

This paper introduces AdDrop, a novel dropout approach which chooses only high-attribution components of attention to apply dropout to. This paper presents experiments on the GLUE benchmark using BERT base and RoBERTa base, and does an excellent job of presenting experiments, ablations, and hyperparameter tuning to showcase the positives and negatives of the proposed approach.

**Questions:**

I think where you say "drop by random sampling" you mean "drop by uniform sampling"?

**Limitations:**

The authors appropriately listed limitations, though it could be made more explicit that the experimental evidence only covers BERT base and RoBERTa base, so the intended use here likely just covers similar pretrained language models during fine-tuning. Further experiments are necessary to evaluation if this works with other models or other types of data.

**Strengths And Weaknesses:**

Strengths:
This paper has many strengths, including the clarity of the writing, the ablations motivation the different experimental choices, and well-chosen baselines (though there could be more baselines, there are so many variants of dropout now it's not possible to cover all baselines here).
The gains presented in Table 1 are surprisingly consistent. My initial thought was that we should have some presentation of the variance (e.g., from random seed), then I found Table 3 which shows exactly that. Table 3 also shows that the variance in the AD-Drop experiments is lower than in the BERT and RoBERTa experiments, which is a benefit of the approach that the authors could emphasize further.
I really like the hyperparameter sensitivity results in Figure 6, as I was reading I was hoping to see something like this. It's clear that RoBERTa has less benefit, but that there is still a range of q for which performance improves.
The appendix is also great, I love that the authors reported hyperparameters, the size of the GLUE datasets, and more experiments.
The transparency here is great, and I believe this paper has a lot of necessary reproducibility information.

Weaknesses:
It's clear to me that the authors appropriately allocated their limited budget to provide experimental evidence supporting their claims that this approach improves performance, so I don't count this as a negative in my review, but to increase adoption of AD-Drop in the community the authors may consider running experiments with larger models (e.g., BERT large and RoBERTa large are more standard). The results may look different; sometimes increased scale means more regularization is necessary, sometimes it means the models are more stable.
I would recommend having the caption of each figure be descriptive enough that it can be considered stand-alone (without also reading the main body of the paper). For example, the caption to Figure 2 should say the results are for MRPC, and Figure 4 should state what p and q are. It's better to have the main takeaway for each figure in the caption as well ("with cross-tuning" leads to much lower variance and higher performance).

---

> ### Author Response · Authors · 2022-08-02
> **Response to Reviewer XQcd**
>
> We appreciate your valuable comments and try to address your concerns as follows.
>
> * **Q1: Consider running experiments with larger models.**
>     > **A1:** Great suggestion. We conduct the evaluation of AD-DROP with RoBERTa-large on RTE and MRPC datasets. The table below shows the average scores and standard deviations of five random seeds. There are two main observations. First, AD-DROP achieves consistent improvements over the larger RoBERTa model, illustrating that AD-DROP is scalable to large models. Second, compared with the RoBERTa-base on RTE in Table 3, the large model significantly reduces the deviation (from 1.7 to 0.86), suggesting that a larger model size indeed helps to improve the stability. And AD-DROP further improves the performance and reduces the deviation. We will supplement these results in the revised paper.
>
>     > **Table: Testing AD-DROP on a larger model.**
>     | Methods       | MRPC         | RTE           |
>     |---------------|--------------|---------------|
>     | RoBERTa-large | 90.83$\pm$0.75 | 85.99$\pm$0.86  |
>     | +AD-DROP      | **91.62$\pm$0.53** | **88.01$\pm$0.48**  |
>
> * **Q2: Regarding better caption of the figures.**
>     > **A2:** Thanks. We will revise the captions accordingly.
>
> * **Q3: Regarding "drop by uniform sampling".**
>     > **A3:** That’s correct. We will clarify it in the revised version.
>
> * **Q4: Regarding additional experiments on other tasks/data types.**
>     > **A4:** To address this concern, we conduct additional experiments on NER (CoNLL-2003) and Machine Translation (WMT2016) tasks. The results validate the effectiveness and generalizability of AD-DROP. For the details, please refer to the **[General Response](https://openreview.net/forum?id=XYDXL9_2P4&noteId=9Idjo23DzV)**.

---

> > ### Comment · Reviewer_XQcd · 2022-08-02
> > **Nice additional experiments, thanks for addressing points**
> >
> > It looks like you ran additional experiments on five datasets (MRPC, RTE, CoNLL-2003, WMT2016 EN-RO, and WMT2016 TR-EN) with three additional models (RoBERTa-large, ELECTRA-base, and OPUS-MT); that's a lot for a rebuttal.
> >
> > I'll keep my score as Accept. Thanks, I believe your work does improve the paper.

---

### Author Response · Authors · 2022-08-02
**General Response to Reviewers**

We sincerely thank all reviewers for their valuable comments, which are crucial for improving our work.
* **Q: Regarding additional experiments on other tasks.**
    > **A:** Although AD-DROP was evaluated on GLUE tasks, it is generally applicable on other tasks as well. To demonstrate this, we conduct additional experiments of AD-DROP on **NER** (CoNLL-2003) and **Machine Translation** (WMT2016) tasks. The results on the test sets are listed below. Moreover, to verify that AD-DROP can be adapted to other pretrained models, for CoNLL-2003 NER, we choose ELECTRA as the baseline in the additional experiments. For WMT2016, the strong baseline OPUS-MT is chosen. The results show that AD-DROP consistently improves the strong baselines on both NER and Machine Translation tasks.

    > **Table: Test results of AD-DROP on the CoNLL-2003 NER dataset.**
    | Methods       | Accuracy     | F1            |
    |:------------:|:------------:|:-------------:|
    | ELECTRA-base | 97.83 | 91.23 |
    | +AD-DROP     | **97.95** | **91.77**  |

    > **Table: Test results of AD-DROP on the WMT2016 EN-RO and TR-EN datasets. The evaluation metric is BLEU.**
    | Methods   | EN-RO | TR-EN  |
    |:--------:|:-----:|:------:|
    | OPUS-MT  | 26.11 | 23.88  |
    | +AD-DROP | **26.43** |  **23.96**      |

---

### Meta-Review · Area_Chair_VMtd · 2022-08-26

**Recommendation:** Accept
**Confidence:** Less certain

**Metareview:**

The paper proposes a method AD-DROP to drop attention weights in a network to alleviate overfitting.
It randomly sample a set of token positions with respect to attribution score calculated in first pass.
The authors provide a variety of experiments on multiple tasks (SNLI, NER, MT, etc.) showing effectiveness comparing to other methods. The method is slower since it needs a separate pass to calculate attention attribution.



**Award:**

No

---

### Decision · Program_Chairs · 2022-09-14

Accept